# OpenReview forum: "MSCR: Exploring the Vulnerability of LLMs’ Mathematical Reasoning Abilities Using Multi-Source Candidate Replacement"
_ICLR.cc/2026/Conference — ICLR 2026 Poster_

### Official Review · Reviewer_ohXA · 2025-10-26

**Soundness:** 2
**Presentation:** 2
**Contribution:** 2
**Rating:** 6
**Confidence:** 3

**Summary:**

The paper introduces MSCR, an automated adversarial attack method that systematically probes the robustness of LLMs in mathematical reasoning. By leveraging multi-source candidate word replacements (embedding similarity, WordNet, and masked language models), MSCR generates semantically similar perturbations to input questions. Experiments on GSM8K and MATH500 benchmarks show that even single-word changes can drastically reduce LLM accuracy and increase response length, revealing significant vulnerabilities in current models.

**Strengths:**

- Proposes a scalable, automated adversarial attack framework that does not require manual intervention.
- Demonstrates effectiveness across 12 open-source and 2 commercial LLMs, with clear quantitative results.
- Reveals not only accuracy drops but also increased response lengths, highlighting efficiency and robustness issues.
- Very interesting find on how seemingly atomic perturbations are causing the model generation to collapse.

**Weaknesses:**

- Limited Defense Discussion: The paper does not propose or evaluate potential defenses or mitigation strategies for the identified vulnerabilities. Also there is a need for more elaborate analysis and root causing on how single word flips is causing such significant drops in performances.
- More Analysis on perturbation quality : tying this back to the lack of root-cause analysis, it would be great to understand the quality/extent of perturbations in the adversarial examples. Also, analysis on what sort of perturbations cause the most performance degradation (in term of both accuracy and response token lengths)
- Over-emphasis on single word perturbations : would be good to understand what the performance degradation looks like for multiple word flips as well - otherwise it does not provide a complete picture.

The aim here should not be to just point out a gap, but also a more thorough analysis on where this gap comes from, how reproducible it is, and how accurate the testing process is. Lack of these aspects do not affirm confidence in the process. Overall, I would suggest a deeper dive into the issue at hand.

**Questions:**

Addressed in the Weakness section.

---

> ### Author Response · Authors · 2025-11-20
>
> Thank you very much for your careful review of our manuscript and for providing valuable comments. We also greatly appreciate your recognition of our work. Below are our responses to the questions you raised, and we hope they will address your concerns.
>
> **Weakness 1** : Limited Defense Discussion: The paper does not propose or evaluate potential defenses or mitigation strategies for the identified vulnerabilities. Also there is a need for more elaborate analysis and root causing on how single word flips is causing such significant drops in performances.
>
> **Answer 1** :
>
> **Discussion of defense or mitigation methods.**
>
> Since our work primarily focuses on the adversarial attack algorithms for LLMs and investigates the robustness of their mathematical reasoning abilities, we do not discuss defense or mitigation strategies in the paper. However, in this response, we propose a potential mitigation strategy: we have designed a powerful and robust prompt to guide LLMs to detect perturbations and minimize their impact. We also conducted experiments to explore the performance of this mitigation strategy, and the results are shown in the table below. We performed the experiments on the GSM8K dataset (accuracy, %).
>
> | Model  | Original | Attack  | Defense |
> | :----: | :------: | :------: | :------: |
> | DeepSeek-R1-Distill-Qwen-1.5B |  83.09   |  57.54   |  60.65   |
> | Qwen2.5-Math-1.5B-Instruct |  85.90   |  36.01   |  38.82   |
> | Meta-Llama-3-8B-Instruct |  83.02   |  47.69   |  51.85   |
> | gemma-2-9b-it |  90.75   |  53.07   |  57.54   |
>
> From the results above, we can analyze that the carefully designed, powerful, and robust prompt slightly mitigates the performance degradation of LLMs during adversarial attacks on mathematical reasoning. However, the models still experience significant performance drops. We believe that this strong prompt design is a feasible approach for defense or mitigation. In the future, designing even more powerful prompts or incorporating adversarial examples during the training of LLMs can further improve defense and mitigation effectiveness.
>
> **An investigation into the reasons why single-word flips lead to such a significant drop in performance.**
>
> Regarding the significant performance drops caused by a single-word flip, we believe this highlights a core issue in current LLMs when handling mathematical reasoning tasks: the model often does not rely on a true "understanding-reasoning" process to generate answers, but instead heavily depends on the patterns or specific solving templates it has memorized during training. When the input closely matches the “familiar” question patterns, the model can easily apply the corresponding template and provide the correct answer. However, when we make a slight surface perturbation to the question, even if the meaning remains almost unchanged, the model often fails to generalize its previous solving patterns to the new phrasing, leading to reasoning failure and incorrect answers.
>
> For example, in the case we show in the paper, replacing “Josh” with “he”: for human readers, this replacement hardly affects the understanding of the question, and the logical structure and correct answer remain unchanged. But for the model, the phrasing deviates from the “familiar” template it has memorized, and the solving pattern it originally relied on can no longer be directly applied. When the model faces this "slightly altered but essentially similar" new input, it lacks sufficient generalization ability, causing it to deviate from the correct track and ultimately produce an incorrect result. This indirectly confirms the current state of LLMs, where they overly rely on memory and pattern matching, rather than robust reasoning abilities.

---

> ### Author Response · Authors · 2025-11-20
>
> **Question 2** : More Analysis on perturbation quality : tying this back to the lack of root-cause analysis, it would be great to understand the quality/extent of perturbations in the adversarial examples. Also, analysis on what sort of perturbations cause the most performance degradation (in term of both accuracy and response token lengths)
>
> **Answer 2** :
>
> **Explanation of the quality and extent of the perturbations.**
>
> In our work, the quality and extent of the perturbations are key aspects we focus on. First, our perturbations are generated using multi-source candidates, leveraging the model's embedding space, WordNet dictionary, and MLM context predictions. This approach significantly expands the variety of perturbation words. Next, each replacement involves a semantically similar word, but we use a strict filtering mechanism to select high-quality candidates, ensuring that the perturbation does not change the core meaning of the problem. We ensure that the perturbations are as small as possible in our experiments, minimizing the impact on the original problem’s logical structure, thus testing the model’s robustness. These small perturbations can effectively trigger the model’s reasoning errors, causing significant performance degradation in mathematical reasoning tasks.
>
> From the perspective of perturbation quality, we quantitatively assess the effect of each replacement. Specifically, we evaluate the changes in the model's reasoning process after each perturbation, particularly the extension of the reasoning chain and the increase in reasoning errors. For example, in many models, we observe that the reasoning chain length increases after perturbation, and in some cases, the reasoning process becomes more lengthy and confused. This suggests that the model not only makes errors in correctness but also shows clear deviations in the order and structure of the reasoning steps. Additionally, in terms of perturbation extent, our replacement method only perturbs one word at a time, ensuring precision and minimizing impact. We also notice that even a single-word replacement can significantly affect the reasoning path of models.
>
> In summary, the adversarial examples generated by our method have high perturbation quality and low perturbation extent. By precisely controlling the quality and extent of the perturbations, we ensure that each replacement only involves a small adjustment at the word level while maximizing semantic consistency. This fine-grained perturbation strategy effectively triggers the model's reasoning biases. Despite the small extent of the perturbations, they significantly impact the model’s reasoning chain and decision-making process, proving the efficiency and precision of our method in testing model robustness.
>
> **An ablation study and analysis of which types of perturbations lead to the most severe performance degradation.**
>
> We conducted ablation studies on the three perturbation candidate word sources to explore which type of perturbation causes the most significant performance degradation. The results are shown in the two tables below, where the first table presents the performance in terms of accuracy, and the second table shows the performance in terms of response token length. The experiments were conducted on the GSM8K dataset.
>
> | Model  | Original | Attack (ours) | Attack (Cosine) | Attack(WordNet) | Attack(MLM) |
> | :----: | :------: | :------: | :------: | :------: | :------: |
> | DeepSeek-R1-Distill-Qwen-1.5B |  83.09   |  57.54   |  61.79   |  63.68   |  61.56   |
> | Qwen2.5-Math-1.5B-Instruct |  58.90   |  36.01   |  51.48   |  51.40   |  49.13   |
> | Meta-Llama-3-8B-Instruct |  83.02   |  47.69   |  51.93   |  54.06   |  53.68   |
> | gemma-3-4b-it |  94.54   |  63.84   |  72.86   |  66.87   |  70.43   |
> | gemma-2-9b-it |  90.75   |  53.07   |  65.50   |  65.88   |  56.71   |
>
> | Model  | Original | Attack (ours) | Attack (Cosine) | Attack(WordNet) | Attack(MLM) |
> | :----: | :------: | :------: | :------: | :------: | :------: |
> | DeepSeek-R1-Distill-Qwen-1.5B |  1754.3   |  2698.0   |  2715.7   |  2740.3   |  2703.5   |
> | Qwen2.5-Math-1.5B-Instruct |  306.8   |  555.9   |  688.9   |  606.9   |  431.4   |
> | Meta-Llama-3-8B-Instruct |  174.3   |  203.1   |  200.0   |  207.6   |  206.8   |
> | gemma-3-4b-it |  411.8   |  881.7   |  833.5   |  910.3   |  918.2   |
> | gemma-2-9b-it |  174.6   |  190.0   |  194.5   |  199.6   |  185.2   |
>
> From the analysis of the two tables, we find that different models are affected to different degrees by different types of perturbations. In terms of accuracy, the MLM information source usually produces stronger attacks and achieves the best attack performance on 3 out of 5 models. In terms of response token length, the perturbations from the WordNet information source are the strongest, as they lead to longer response token lengths than the other two sources on 3 out of 5 models.

---

> > ### Author Response · Authors · 2025-11-20
> >
> > **Question 3** : Over-emphasis on single word perturbations : would be good to understand what the performance degradation looks like for multiple word flips as well - otherwise it does not provide a complete picture.
> >
> > **Answer 3** : We intentionally place a strong emphasis on single-word perturbations and restrict the number of replaced words to one, because we want to ensure that the semantics of the question remain essentially unchanged before and after the modification. However, to address your concerns, we also conduct an additional ablation study to investigate how model performance degrades when perturbing multiple words. Specifically, we choose to perturb two words at the same time, meaning that in the modified question, two words are different from the original question. The results are shown in the table below, and this ablation study is conducted on the GSM8K (accuracy, %).
> >
> > | Model  | Original | Attack(one word) | Attack(two word) |
> > | :----: | :------: | :------: | :------: |
> > | DeepSeek-R1-Distill-Qwen-1.5B |  83.09   |  57.54   |  51.86   |
> > | Qwen2.5-Math-1.5B-Instruct |  85.90   |  36.01   |  26.76   |
> > | Meta-Llama-3-8B-Instruct |  83.02   |  47.69   |  40.03   |
> > | gemma-3-4b-it |  94.54   |  63.84   |  55.42   |
> > | gemma-2-9b-it |  90.75   |  53.07   |  47.08   |
> >
> > The results show that perturbing multiple words can indeed improve the attack effectiveness. However, here we want to emphasize that multi-word perturbation is not the main intention of our work. Our goal is to make the model answer incorrectly by changing only a single word, while keeping the overall meaning of the problem essentially unchanged. When the number of perturbed words increases, the meaning of the modified problem can deviate too much from the original one, so it becomes natural for the model to make mistakes. This is no longer aligned with our central idea of causing model failures under minimal, semantics-preserving perturbations.

---

> > > ### Author Response · Authors · 2025-11-27
> > >
> > > Dear Reviewer:
> > >
> > > We hope we haven't bothered you. With the discussion deadline fast approaching, we would like to know whether we have adequately addressed your concerns.
> > >
> > > Every piece of feedback you provide is invaluable to us and crucial for improving the quality of our research. If you have any further questions, we would be happy to continue to answer and revise your work.
> > >
> > > Thank you sincerely for taking the time out of your busy schedule to review our paper and responses.

---

### Official Review · Reviewer_EQZP · 2025-10-29

**Soundness:** 3
**Presentation:** 3
**Contribution:** 2
**Rating:** 4
**Confidence:** 5

**Summary:**

The vulnerability of LLMs in mathematical reasoning is a significant area of research. This work introduces an automatic adversarial attack method that generates perturbations by replacing words with candidates from three information sources.  The method can successfully reduce the accuracy of all models by altering just a single word. The authors also discovered that these perturbations cause a substantial increase in the average length of the model's responses.

**Strengths:**

The paper presents a valuable contribution by proposing an automatic method to study the vulnerability of LLMs in mathematical reasoning. The methodology is clearly explained, and the results appear reasonable.  The finding regarding response length is interesting, which offers a novel insight into model robustness beyond simple accuracy.

**Weaknesses:**

The main performance results do not offer new insights beyond established literature, making the method's added value unclear.

Key analyses are absent, including an ablation study on the role of the three information sources and any metrics regarding the attack's efficiency (e.g., the number of candidate replacements needed for success).

**Questions:**

Please clarify whether the measured response length for models like the DeepSeek-R1-Distill-Qwen series includes their long chain-of-thought tokens. If not, a discussion of whether the adversarial attacks had a differential impact on the long cot could reveal insights into their relative robustness.

---

> ### Author Response · Authors · 2025-11-20
>
> We sincerely thank you for taking the time to review our manuscript and for patiently providing valuable comments and suggestions on our paper. Below are our responses to the issues and concerns you raised, and we hope they address your questions.
>
> **Weakness 1** : The main performance results do not offer new insights beyond established literature, making the method's added value unclear.
>
> **Answer 1** : We address the concern about the method’s added value from both the methodological side and the empirical findings.
>
> First, at the method level, existing text adversarial attack methods such as TextFooler and BERT-ATTACK mainly target discriminative models (e.g., text classification, natural language inference) and BERT-style masked language models. Their primary goal is to make a fine-tuned classifier mispredict while keeping the sentence fluent. Thus, their attack space, evaluation metrics, and application scenarios are fundamentally different from those of today’s generative large language models. They cannot be directly applied to the setting we focus on, namely long-form generation with chain-of-thought reasoning for math problems. In contrast, our method MSCR is specifically designed as a single-word, automatic attack framework for generative LLMs. It interacts with reasoning models in a black-box way, uses multi-source candidates to construct high–semantic-consistency substitutions, and generates adversarial examples at scale. This is fundamentally different from traditional classifier-oriented attacks like TextFooler、BERT-ATTACK in terms of the attack target and setting.
>
> Second, in the literature on safety and robustness for generative LLMs, many works focus on jailbreak attacks, for example by optimizing prompt suffixes to bypass alignment and make the model output harmful or unsafe content. Their evaluation dimension is mainly safety/alignment, not mathematical reasoning ability, and they are usually tested on general open-domain tasks or a few safety benchmarks. These methods are not directly suitable for a systematic evaluation of math reasoning robustness. Our work is different: on the one hand, the attack target is the math reasoning task itself, and we systematically cover the GSM8K and MATH500 math datasets and 12 open-source reasoning models and 2 commercial models. On the other hand, our method is fully automatic and does not require manual prompt or rule design. Given any math problem and any generative LLM, our pipeline can directly run and produce adversarial question–answer pairs. This is clearly different from benchmarks that require humans to construct distracting sentences or templates, which is costly and hard to scale to more datasets and models.
>
> More importantly, at the level of empirical findings, we show a new type of vulnerability under an extremely small perturbation regime: only one word is changed, under a strict constraint of semantic preservation. Under this setting, we systematically demonstrate a new vulnerability pattern in current reasoning LLMs on math tasks: substituting a single ordinary word, sometimes just a proper name, pronoun, or conjunction, while keeping the mathematical structure and semantics unchanged for humans, can still cause a large drop in accuracy across many models, and is accompanied by a clear “reasoning length collapse”. For modified questions, the model’s response length is often about 1.5–2× the original on average, and in extreme cases up to 10×. This shows that under very small linguistic perturbations, models shift from a concise and stable reasoning mode to a much longer and less stable reasoning mode, so that robustness and reasoning cost degrade at the same time. Prior works such as GSM-IC, GSM-Plus, GSM-Symbolic, and MATH-Perturb mainly analyze robustness through accuracy/error rates and do not provide a systematic characterization of how reasoning patterns change together with computational cost.
>
> Taken together, we argue that our work has clear added value.  We propose a single-word, automatic attack framework for generative LLMs that is low-cost and easily reusable, filling the gap left by methods like TextFooler and BERT-ATTACK in the generative reasoning setting.  We bring adversarial attacks from generic safety jailbreak scenarios into the high-value domain of math reasoning, and show that our method generalizes well across different math datasets and a wide range of models, including commercial ones.  Under minimal, semantics-preserving perturbations, we reveal a joint collapse of reasoning accuracy and reasoning cost, which refines the existing understanding of robustness in the literature from a new perspective. Overall, our method provides unique value, offers important support for future robustness studies, and gives useful insights for improving the reasoning abilities of LLMs.

---

> > ### Author Response · Authors · 2025-11-20
> >
> > **Weakness 2** : Key analyses are absent, including an ablation study on the role of the three information sources and any metrics regarding the attack's efficiency (e.g., the number of candidate replacements needed for success).
> >
> >
> > **Answer 2** : We conduct an ablation study to show the attack performance when using each of the three information sources separately. The results are shown in the following figure. The ablation study is conducted on the GSM8K (accuracy, %).
> >
> > | Model  | Original | Attack (ours) | Attack (Cosine) | Attack(WordNet) | Attack(MLM) |
> > | :----: | :------: | :------: | :------: | :------: | :------: |
> > | DeepSeek-R1-Distill-Qwen-1.5B |  83.09   |  57.54   |  61.79   |  63.68   |  61.56   |
> > | Qwen2.5-Math-1.5B-Instruct |  58.90   |  36.01   |  51.48   |  51.40   |  49.13   |
> > | Meta-Llama-3-8B-Instruct |  83.02   |  47.69   |  51.93   |  54.06   |  53.68   |
> > | gemma-3-4b-it |  94.54   |  63.84   |  72.86   |  66.87   |  70.43   |
> > | gemma-2-9b-it |  90.75   |  53.07   |  65.50   |  65.88   |  56.71   |
> >
> > Our experimental results show that our attack method significantly outperforms attacks using only a single information source in terms of model performance degradation. Specifically, on all models, the attack using the combination of three information sources performs better than using any single source alone. At the same time, the attack performance of WordNet is often lower than that of Cosine similarity and MLM, but the combination of all three tends to achieve more impressive results. These results suggest that combining multiple information sources improves the effectiveness of adversarial attacks, and the performance of each source varies across different models. Our ablation study further verifies the importance of the three information sources in constructing high-quality adversarial samples, providing a quantitative analysis of the contribution of each information source.
> >
> > Regarding attack efficiency, we present the substitution count metric in the table below, which reflects the average number of substitutions required for different models to successfully attack the GSM8K and MATH500 datasets.
> >
> > | Model   | GSM8K | MATH500 |
> > | :----: | :-----: | :-----: |
> > | DeepSeek-R1-Distill-Qwen-32B  | 37.56   | 12.86   |
> > | DeepSeek-R1-Distill-Qwen-7B  | 21.76   | 12.51   |
> > | DeepSeek-R1-Distill-Qwen-1.5B  | 7.00   | 5.59   |
> > | Qwen2.5-Math-7B-Instruct  | 19.77   | 11.69   |
> > | Qwen2.5-Math-1.5B-Instruct  | 10.60   | 9.36   |
> > | Meta-Llama-3-70B-Instruct  | 46.05   | 9.16   |
> > | Meta-Llama-3-8B-Instruct  | 14.17   | 2.41   |
> > | gemma-3-27b-it  | 42.47   | 16.54   |
> > | gemma-3-12b-it  | 38.36   | 13.73   |
> > | gemma-3-4b-it  | 13.94   | 7.67   |
> > | gemma-2-27b-it  | 28.31   | 11.59   |
> > | gemma-2-9b-it  | 12.71   | 7.86   |
> >
> > Overall, our attack algorithm shows good efficiency: the average number of substitutions needed for a successful attack is relatively small. Although some large-parameter models require more substitutions, our method still has a clear efficiency advantage. First, it is fully automatic and does not require human intervention, and it maintains good generalization across different models and datasets. Unlike many other adversarial attack methods, our algorithm does not rely on hand-crafted rules or heavy manual effort; instead, it runs efficiently in a fully automated framework, which saves both computational and human cost. Therefore, our method is not only efficient, but also highly adaptable and scalable. In summary, our algorithm can automatically generate adversarial examples while still achieving a high success rate and consistent performance across multiple models and datasets. This efficient and widely applicable attack framework can greatly support future robustness research and provide important insights for improving the reasoning ability of LLMs.

---

> > > ### Author Response · Authors · 2025-11-20
> > >
> > > **Question 1** : Please clarify whether the measured response length for models like the DeepSeek-R1-Distill-Qwen series includes their long chain-of-thought tokens. If not, a discussion of whether the adversarial attacks had a differential impact on the long cot could reveal insights into their relative robustness.
> > >
> > > **Answer 1** : Thank you for raising this question. You are right that we do not explicitly state in the paper whether the measured response length for the DeepSeek-R1-Distill-Qwen series includes their long chain-of-thought (CoT) tokens. Here we clarify that our measured response length does include the CoT tokens for these models. We do this because, for reasoning-style models like DeepSeek-R1-Distill-Qwen, most of the computational cost and inference overhead comes from the generated reasoning steps in the chain-of-thought, so it is natural to count these tokens when we measure response length.
> > >
> > > We also observe that, for the DeepSeek-R1-Distill-Qwen series, the increase in response length under adversarially perturbed questions mainly comes from longer chains-of-thought. In these cases, the model’s reasoning process becomes longer and more complex, and we often see signs of “getting stuck,” such as repetitive reasoning or confusion. This means that, when a reasoning-capable LLM faces our substitutions, it not only changes its final numerical answer, but also shows a strongly altered and less stable reasoning path, which leads to a longer reasoning chain. Therefore, the growth in CoT tokens is a significant part of the impact of the adversarial attacks on these models’ robustness, and it clearly reflects an increase in the computational cost of the reasoning process.
> > >
> > > Thank you very much for pointing out this issue. We have added the relevant content in Section 4 of the revised version of our paper.

---

> > > > ### Author Response · Authors · 2025-11-27
> > > >
> > > > Dear Reviewer,
> > > >
> > > > We apologize for the intrusion. With less than a week remaining in the discussion period, we would like to confirm whether you are satisfied with our previous response, or whether your question has been fully resolved.
> > > >
> > > > Please let us know if there is anything further that requires our clarification. Your feedback is invaluable to us, and we look forward to refining our work based on your comments.
> > > >
> > > > Thank you again for your time and review.

---

### Official Review · Reviewer_ZoDp · 2025-10-29

**Soundness:** 2
**Presentation:** 2
**Contribution:** 1
**Rating:** 2
**Confidence:** 5

**Summary:**

This work proposes an automated adversarial attack framework, MSCR, which uses pre-existing external information sources and LLM embedding spaces to identify semantically coherent replacements for a single word in an input mathematical problem. Use of this framework demonstrates that even a single word, semantically similar replacement that is non-task-specific (irrelevant to the mathematical information) is a perturbation significant enough to cause reasoning failures, even in frontier models. This work also demonstrates collateral liabilities - namely the longer (incorrect) responses, and associated higher computational resource consumption.

**Strengths:**

1. This work presents a useful analysis framework by demonstrating the distribution of response lengths and increase resource consumption, beyond the reduction in accuracy of output
2. This work effectively demonstrates how even minor, non-task-specific perturbations cause a significant decline in model performance

**Weaknesses:**

1. The observations and conclusions drawn in this work are not novel. The following works have established more substantial claims and contributions in the domain of adversarial input generation, including for mathematical problems on GSM8K
- Cutting through the Noise Anantheswaran et al. (2025)
- MathAttack Zhou et al. (2024)
- Adversarial Math Word Problem Generation Xie et al. (2024)
- An LLM can Fool Itself Xu et al. (2023)
- Universal and Transferable Adversarial Attacks on Aligned Language Models Zou et al. (2023)
- Evaluating Models’ Local Decision Boundaries via Contrast Sets Gardner et al. (2020)

The adversarial attacks in this work are of a lower complexity and frequency than the related works. While this work demonstrates that even minor perturbations can cause model failures, the framework proposed is not novel or usable for high complexity adversarial attacks
2. Fig 1: The example demonstrates the effect of a single word replacement. However, the work fails to detail
- 2.1 how the substitution of semantically similar candidates makes for better adversaries than random replacement
- 2.2 why, in this case, the substitution of a generic term ("he") in the absence of a strongly semantically-related term is effective
3. Why the substitution causes model reasoning failures is not adequately explored or hypothesized on - especially since the perturbations are not specific to the mathematical task.

**Questions:**

1. Is there a causal link or a hypothesis that could explain the effect of the substitutions on the model reasoning patterns?
2. This work could benefit from ablation studies exploring the effects of replacing more than one word; replacing words with candidates that have a cosine similarity above a certain threshold (thus ensuring a strong semantic similarity, not just the closest semantically similar word); the effects of replacing different words or sentence components for the same input and whether the choice of word to replace can offer explanations as to why the model reasoning is derailed by these minor perturbations.
3. This work could also benefit from conducting the same inference experiments on LLMs such as GPT/Claude/Gemini, even if on a smaller subset.

---

> ### Author Response · Authors · 2025-11-20
>
> We are very grateful that you took the time to review our manuscript and to offer your valuable comments and suggestions. Your advice plays a crucial role in improving the quality of our paper. Below are our responses to the weaknesses and questions you raised regarding our work, and we hope they address your concerns.
>
> **Weakness 1** : An inquiry about the lack of novelty in this work.
>
> **Answer 1** : We do not agree with your assessment that the observations and conclusions in this work are not novel. While the papers you listed each make important contributions, they fundamentally differ from our research in several key aspects.
>
> First, *Cutting Through the Noise: Boosting LLM Performance on Math Word Problems* focuses on how LLMs perform on math word problems with irrelevant information, where the perturbations mainly add additive noise in the form of unrelated variables and noise sentences. In contrast, our MSCR focuses on single-word perturbations with an emphasis on minimal modification while preserving semantic consistency. *MathAttack: Attacking Large Language Models Towards Math Solving Ability* also uses substitution, but its attacks mainly target ordinary words and does not systematically discuss the effect of substituting special logical entities like proper names. For example, the replacement of “Josh” with “he” in our paper does not change the meaning but causes significant reasoning failures, a phenomenon not covered in MathAttack. As for *Adversarial Math Word Problem Generation*, it generates new problems by modifying numbers and reconstructing abstract syntax trees, which changes the numerical structure and truth values of the original problem. While the difficulty and form may remain similar for humans, the mathematical facts differ. This differs from our adversarial threat model, where we require the logical structure and answers to remain unchanged for humans. We focus on the local robustness collapse of the model under the constraint of unchanged tasks and answers, rather than generating entirely new problems. Moreover, these works are mainly focused on the GSM8K and cannot be directly extended to more challenging mathematical reasoning tasks like MATH500, whereas our experiments cover both GSM8K and the more challenging MATH500, demonstrating that MSCR applies not only to simpler application problems but also to more complex mathematical reasoning tasks.
>
> For *An LLM Can Fool Itself: A Prompt-Based Adversarial Attack* and *Universal and Transferable Adversarial Attacks on Aligned Language Models*, these are typical prompt-based jailbreak approaches that optimize prompt suffixes to make models output unsafe content on limited harmful datasets. Their evaluation focuses on safety and alignment, rather than the correctness of mathematical reasoning and reasoning path structures, which is different from our focus on mathematical reasoning robustness. *Evaluating Models’ Local Decision Boundaries via Contrast Sets* constructs contrast sets through manual processes to study local decision boundaries in traditional NLP models. However, it is not targeted at large-scale CoT mathematical reasoning scenarios and relies heavily on manual construction, making it much more costly than our MSCR framework, which is fully automated, requires no human intervention, and can scale to multiple models and datasets at low cost.
>
> The key point is that our MSCR requires only a single-word substitution to generate high-quality adversarial examples. The value of our method is not in creating high-complexity attacks with substantial semantic rewrites, structural rearrangements, or numerical changes, but in intentionally reducing the attack budget to minimal word-level perturbations, examining the model’s local decision boundaries’ vulnerability when semantic changes are almost negligible. The fact that such small perturbations can still cause significant accuracy drops and increased reasoning costs across multiple models and datasets highlights a serious limitation in current LLMs' ability to handle small linguistic changes in mathematical reasoning.
>
> Finally, our observations and conclusions are also novel. Most of the works you mentioned focus primarily on metrics such as accuracy changes or attack success rates, with limited exploration of how the reasoning path itself is affected. Our work not only confirms the significant performance drops but also systematically reveals how the length of the reasoning chain increases significantly under minimal linguistic perturbations, with the reasoning process shifting from a concise and stable mode to a longer, repetitive, or even contradictory mode, leading to the so-called "length collapse" and a sharp increase in reasoning costs. This dual degradation has not been explored in the papers you mentioned. Therefore, we believe that our work is both novel and highly valuable, whether in terms of attack design, scope of models and datasets, or analysis perspective.

---

> > ### Author Response · Authors · 2025-11-20
> >
> > **Weakness 2.1** : how the substitution of semantically similar candidates makes for better adversaries than random replacement.
> >
> > **Answer 2.1** : We first want to clarify that our goal is to perform adversarial attacks, not to simply “mess up” the problem. In text adversarial attacks, a core and widely accepted principle is that we should keep the original meaning and task unchanged for humans as much as possible, while making the model’s output incorrect. In other words, a good adversarial example should still be judged by a human as having the same meaning as the original sample, but the model fails because of a small change. Based on this principle, we use substitution of semantically similar candidates to construct adversaries, so that the perturbed question still looks like the same question to a human, and we can modify the problem without changing its intended meaning. This allows us to create stronger and more meaningful adversarial examples. In contrast, if we use random replacement, it is very likely to change key entities, numbers, relations, or even the whole intent of the question, producing inputs whose semantics are corrupted or no longer reasonable. This is closer to data poisoning than to a true adversarial example. Such random replacement can of course make the model wrong, but it moves away from the original goal of adversarial attack, so we do not consider it a suitable or valid approach in our setting.
> >
> >
> > **Weakness 2.2** : why, in this case, the substitution of a generic term ("he") in the absence of a strongly semantically-related term is effective.
> >
> > **Answer 2.2** ：We believe this phenomenon clearly exposes a core issue of current LLMs in mathematical reasoning: the model largely relies on memory and template-like pattern matching, instead of doing truly generalizable reasoning. In the original problem, the name "Josh" appears, and this kind of pattern is very frequent in training data and math word problems. The model is likely to build a very stable solution pattern for this type of input: once it sees this surface pattern, it directly activates a familiar chain-of-thought template. When we substitute "Josh" with the generic term "he", humans can easily use the context to link "he" back to the same person, so the mathematical structure and the essential meaning of the problem stay the same. But for the model, this substitution breaks the familiar surface pattern. It can no longer just reuse a name-based template and instead needs to understand the deeper structure and meaning of the text. If the model does not handle this step well, its internal representation shifts, and the entire reasoning chain is more likely to fail.
> >
> > In other words, in this example, "he" and "Josh" may not be a strong pair in a static word-level semantic space, but in the concrete context they play the same semantic role, and "he" clearly refers back to "Josh". Our adversarial attack exactly uses this: for a human, this is still a meaning-preserving, task-preserving minimal perturbation; for the model, the substitution cuts off its original memory and template path and forces it to perform real coreference and structural understanding, where its generalization ability is weaker, so it fails. This example shows that even if the substitute is not the most semantically similar token in an embedding space, as long as it is a contextually valid coreferential substitution, it can still be a high-quality and strong adversarial example.
> >
> > It is important to stress that this does not mean that only "proper name → pronoun" substitutions are effective, nor that our method purposely searches for less semantically related candidates. This case is just one representative example among many successful attacks. In the paper, we also show other types of perturbations that cause clear performance drops while keeping the problem’s meaning unchanged for humans. Taken together, these examples show that current models rely heavily on specific phrasings and surface cues in math problems, and once they are pushed away from these patterns, their reasoning paths tend to break down. This further supports our claim that contextually meaningful substitutions produce more informative adversaries than random replacement and better reveal the internal weaknesses of LLMs.

---

> > > ### Author Response · Authors · 2025-11-20
> > >
> > > **Weakness 3** : Why the substitution causes model reasoning failures is not adequately explored or hypothesized on - especially since the perturbations are not specific to the mathematical task.
> > >
> > > **Answer 3** : We believe that the reason why the substitution of non–math-specific words can still cause severe model reasoning failures is that current LLMs do not truly learn deep understanding and reasoning. Instead, they mainly learn the distribution of the most likely next text given the context. As a result, when solving math word problems, the model often relies on surface patterns and templates it has seen during training, such as specific phrasings, sentence structures, and coreference patterns, and then uses these patterns to trigger a standard solution trajectory. Once we slightly change these surface forms at the single-word level, even if the meaning remains almost equivalent for human readers, the input may move from a high-frequency and familiar pattern region to a slightly out-of-distribution region for the model. This can cause the model to switch to a less reliable reasoning path, which then leads to errors in the intermediate calculation steps and in the final answer, even though the perturbations are not directly tied to the mathematical content itself.
> > >
> > > **Question 1** : Is there a causal link or a hypothesis that could explain the effect of the substitutions on the model reasoning patterns?
> > >
> > > **Answer 1** : We believe there is a clear hypothesis that explains the causal link between the substitutions and the changes in the model reasoning patterns. Our view is that single-word substitutions change the semantic distribution of the input and thereby break the model’s internal mapping from natural language descriptions to abstract mathematical structures, which in turn alters its reasoning patterns. More concretely, when solving math problems, large language models usually rely on surface patterns learned during training (such as common phrasings, sentence structures, and coreference patterns) to construct a reasoning path. Our method MSCR replaces just one word and thus changes these familiar patterns, causing the model to make small mistakes already in the early semantic understanding stage. This early deviation leads the model to build an incorrect internal mathematical structure, which then affects the entire downstream reasoning path.
> > >
> > > Furthermore, since the model relies heavily on statistical patterns during reasoning, even a small change after substitution can push it onto a wrong trajectory, often causing it to produce a longer and more chaotic chain-of-thought, which increases the probability of cumulative errors. In other words, although the substitution does not directly change the underlying mathematics of the problem, it disrupts the mapping that the model uses to go from language to mathematical structure when performing mathematical reasoning, and this is what leads to the reasoning failures. Therefore, the effect of the substitutions is not only at the language surface level, but also at the level of model reasoning patterns and reasoning paths, resulting in a significant drop in mathematical reasoning performance.

---

> > > > ### Author Response · Authors · 2025-11-20
> > > >
> > > > **Question 2** : This work could benefit from ablation studies exploring the effects of replacing more than one word; replacing words with candidates that have a cosine similarity above a certain threshold (thus ensuring a strong semantic similarity, not just the closest semantically similar word); the effects of replacing different words or sentence components for the same input and whether the choice of word to replace can offer explanations as to why the model reasoning is derailed by these minor perturbations.
> > > >
> > > > **Question 2.1** : An inquiry about ablation studies on the replacement of multi-word word.
> > > >
> > > > **Answer 2.1** : To study the effects of replacing more than one word, we run ablation studies where we perturb two words at the same time, i.e., the final perturbed question differs from the original question in two word positions. The results are shown in the table below. Ablation studies are conducted on the GSM8K dataset (accuracy, %).
> > > >
> > > > | Model  | Original | Attack(one word) | Attack(two word) |
> > > > | :----: | :------: | :------: | :------: |
> > > > | DeepSeek-R1-Distill-Qwen-1.5B |  83.09   |  57.54   |  51.86   |
> > > > | Qwen2.5-Math-1.5B-Instruct |  85.90   |  36.01   |  26.76   |
> > > > | Meta-Llama-3-8B-Instruct |  83.02   |  47.69   |  40.03   |
> > > > | gemma-3-4b-it |  94.54   |  63.84   |  55.42   |
> > > > | gemma-2-9b-it |  90.75   |  53.07   |  47.08   |
> > > >
> > > > As the results in the table show, perturbing multiple words does increase the attack effectiveness, and the attack success improves to some extent for each model. However, we do not consider multi-word perturbation to be a feasible direction. Compared with our single-word setting, the gain in effectiveness is not substantial, and changing multiple words makes the semantics of the modified question deviate too much from the original one. This does not align with the main idea of our paper, which is to make the model fail without changing the original meaning of the problem.
> > > >
> > > > **Question 2.2** : An inquiry about an ablation study that investigates the effect of selecting replacement candidates by filtering them based on a cosine similarity threshold.
> > > >
> > > > **Answer2.2** : Because we find that the cosine similarity between the candidate replacement words generated by each information source and the original words is inconsistent and varies greatly in our experiments, our method does not use a fixed cosine similarity threshold for filtering. Instead, it extracts the top few candidate words with the highest similarity from each information source. However, regarding your suggestion to replace words with candidate words whose cosine similarity is above a certain threshold, we also conduct an ablation study. We change the cosine similarity threshold, choosing 0.9, 0.85, 0.8, and 0.75 respectively, and the results are shown in the table below. The ablation study is conducted on the GSM8K dataset (accuracy, %).
> > > >
> > > > | Model  | Original | Attack (ours) | Attack (0.75) | Attack (0.80) | Attack (0.85) | Attack (0.9) |
> > > > | :----: | :------: | :------: | :------: | :------: | :------: | :------: |
> > > > | DeepSeek-R1-Distill-Qwen-1.5B |  83.09   |  57.54   |  58.68   |  60.65   |  62.24   |  63.15   |
> > > > | Qwen2.5-Math-1.5B-Instruct |  85.90   |  36.01   |  36.39   |  37.83   |  41.85   |  47.99   |
> > > > | Meta-Llama-3-8B-Instruct |  83.02   |  47.69   |  48.22   |  48.82   |  49.20   |  51.63   |
> > > >
> > > > From the table above, we can see that as the cosine similarity threshold increases, the attack effectiveness becomes worse. This is because a higher threshold reduces the number of available candidate replacement words, which in turn weakens the attack. At the same time, we find that even when we lower the threshold to 0.75, the performance still does not match our original filtering mechanism. The reason is that relying too much on a cosine similarity threshold filters out some candidates that are not very similar in embedding space but still do not change the original meaning after substitution, for example the case of “Josh” and “he” discussed above. In contrast, the filtering mechanism used in our paper reduces the chance of excluding such useful substitutions.

---

> > > > > ### Author Response · Authors · 2025-11-20
> > > > >
> > > > > **Question 2.3** : An inquiry about the effects of applying different perturbations to words within the same input, and an investigation into the reasons for the model’s reasoning failure as a result of these minor perturbations.
> > > > >
> > > > > **Answer 2.3** : We study the case where we replace the same word in an input with different perturbation words, and we find that this leads to different reasoning results. As shown in the example below, when we replace “try” with “trying” or with “TRY”, the model produces different chains-of-thought and different answers. This difference in the choice of candidate substitutions further explains why small perturbations can disrupt the model’s reasoning: the model is often not truly solving the problem by reasoning, but by matching learned solution patterns. When a small perturbation breaks the original pattern the model relies on, the model is pushed into a pattern it has never really seen during training, which then causes its reasoning to fail.
> > > > >
> > > > > Original question:
> > > > >
> > > > > *Josh decides to try flipping a house.  He buys a house for \\$80,000 and then puts in \\$50,000 in repairs.  This increased the value of the house by 150%.  How much profit did he make?*
> > > > >
> > > > > The model's response when "try" is replaced with "trying":
> > > > >
> > > > > *To determine the profit Josh made, we need to follow these steps:*
> > > > >
> > > > > *1. Calculate the total cost of the house after repairs.*
> > > > >
> > > > > *2. Determine the increased value of the house due to repairs.*
> > > > >
> > > > > *3. Calculate the new value of the house.*
> > > > >
> > > > > *4. Subtract the total cost from the new value to find the profit.*
> > > > >
> > > > > *First, let's find the total cost of the house after the repairs. The initial cost of the house is \\$80,000 and Josh spends an additional \\$50,000 on repairs. So, the total cost is: 80,000 + 50,000 = 130,000*
> > > > >
> > > > > *Next, we need to determine the increased value of the house due to the repairs. The repairs increased the value of the house by 150%. This means the increased value is 150% of the cost of the repairs. We can calculate this as follows: 1.5 * 50,000 = 75,000*
> > > > >
> > > > > *Now, we add this increased value to the initial cost of the house to find the new value of the house: 80,000 + 75,000 = 155,000*
> > > > >
> > > > > *Finally, we subtract the total cost from the new value to find the profit: 155,000 - 130,000 = 25,000
> > > > > Therefore, the profit Josh made is:  \boxed{25000}*
> > > > >
> > > > >
> > > > > The model's response when "try" is replaced with "TRY":
> > > > >
> > > > > *To determine Josh's profit, we need to follow these steps:*
> > > > >
> > > > > *1. Calculate the increased value of the house after repairs.*
> > > > >
> > > > > *2. Determine the total value of the house after the repairs.*
> > > > >
> > > > > *3. Subtract the original cost of the house and the cost of repairs from the total value to find the profit.*
> > > > >
> > > > > *First, let's calculate the increased value of the house after repairs. The original value of the house is \\$80,000, and the repairs increased the value by 150%. An increase of 150% means the value is multiplied by 1.5.*
> > > > >
> > > > > *The increased value is: 80,000 * 1.5 = 120,000*
> > > > >
> > > > > *Next, we add the cost of the repairs to the increased value to get the total value of the house: 120,000 + 50,000 = 170,000*
> > > > >
> > > > > *Now, we subtract the original cost of the house and the cost of the repairs from the total value to find the profit: 170,000 - 80,000 - 50,000 = 40,000*
> > > > >
> > > > > *Therefore, Josh's profit is: \boxed{40000}*
> > > > >
> > > > >
> > > > > **Question 3** : This work could also benefit from conducting the same inference experiments on LLMs such as GPT/Claude/Gemini, even if on a smaller subset.
> > > > >
> > > > > **Answer 3** : Our method relies on cosine similarity over the model’s embedding space to find semantically similar candidate words, which requires direct access to the LLM’s embedding layer. This is only possible for open-source models. However, as far as we know, GPT, Claude, and Gemini are closed-source LLMs, and their internal embeddings are not accessible, so we cannot directly apply our substitution attack pipeline to these models in the same way. Instead, we perform transfer experiments, as shown in Table 2 on page 8 of our paper: we first generate adversarial examples using an open-source model (gemma-3-27b-it), and then run the same inference experiments on commercial models such as o3 and GPT-4o using these fixed adversarial inputs. These transfer results also show strong degradation in performance, which suggests that our adversarial examples generalize well to these models. For your convenience, we include the corresponding results again in the table below (accuracy, %).
> > > > >
> > > > > | Model  | GSM8K (ori) | GSM8K (att) | MATH500 (ori) | MATH500 (att) |
> > > > > | :----: | :------: | :------: | :------: | :------: |
> > > > > | OpenAI-o3 |  97.41   |  81.53   |  98.20   |  91.40   |
> > > > > | GPT-4o |  96.28   |  79.07   |  76.00   |  68.20   |

---

> > > > > > ### Author Response · Authors · 2025-11-27
> > > > > >
> > > > > > Dear Reviewer:
> > > > > >
> > > > > > We hope this message has not been inconvenient. With the discussion deadline approaching, we would like to take the liberty of confirming that we have adequately addressed all your questions.
> > > > > >
> > > > > > If you have any further questions or suggestions, we would love to hear from you. Your feedback is crucial to our work, and we sincerely hope to resolve all your concerns.
> > > > > >
> > > > > > Thank you again for your valuable time in reviewing our paper and response.

---

### Official Review · Reviewer_5XZn · 2025-11-01

**Soundness:** 3
**Presentation:** 3
**Contribution:** 3
**Rating:** 6
**Confidence:** 3

**Summary:**

This paper introduces MSCR (Multi-Source Candidate Replacement), an automated adversarial attack method designed to evaluate the robustness of Large Language Models (LLMs) on mathematical reasoning tasks. The core idea is to generate semantically similar, single-word perturbations for input questions and observe the impact on model performance. MSCR creates a set of candidate replacement words from three sources: cosine similarity in the model's embedding space, the WordNet dictionary, and contextual predictions from a masked language model. The authors conduct extensive experiments on 12 open-source LLMs and test the transferability of the generated attacks on 2 commercial LLMs (OpenAI-03 and GPT-4o), using the GSM8K and MATH500 benchmarks. The main contributions are:
An effective, automated attack framework (MSCR) that requires no manual effort and can systematically probe LLM vulnerabilities in mathematical reasoning. Demonstration of significant robustness failures: The study shows that even a single-word perturbation can cause drastic accuracy drops across all tested model. Adversarial examples generated for one model are shown to be effective against other models, including powerful commercial ones like GPT-4o. The paper reveals that under attack, LLMs not only produce incorrect answers but also generate significantly longer responses. This phenomenon, termed "response length collapse" (though it's more of an expansion), indicates that models enter more complex and inefficient reasoning paths when confused, increasing computational costs.

**Strengths:**

* The study's strength lies in its evaluation across 12 diverse open-source LLMs and 2 commercial models, providing a broad and picture of the vulnerability.
* MSCR is a practical and scalable method. By being fully automated, it overcomes the cost and scalability limitations of manual prompt engineering, making rigorous robustness testing more accessible.
* The discovery that adversarial perturbations lead to significantly longer and more convoluted reasoning paths is a key strength. This "response length collapse" finding connects robustness to efficiency, revealing a hidden cost of model failure that has been less explored.

**Weaknesses:**

* The paper excels at demonstrating that the attacks are successful but offers limited insight into why they work. An analysis of which candidate source (embeddings, WordNet, MLM) generates the most effective perturbations, or what grammatical types of words (e.g., proper nouns, prepositions, verbs) are most vulnerable, would add more insights.
* The use of qwen3-max-preview for secondary evaluation introduces a potential confounder. The results are implicitly dependent on the reliability and potential biases of this specific model.

**Questions:**

1. Could you provide a more detailed breakdown of the successful attacks? Specifically, what percentage of successful perturbations originated from each of the three candidate sources (cosine similarity, WordNet, and MLM)? This would help in understanding which type of semantic similarity LLMs are most sensitive to.
2. Following up on the analysis of attack mechanisms, have you observed any patterns in the types of words that are most critical to perturb? For instance, are proper nouns, which require grounding, more vulnerable than common nouns or verbs? The provided examples are excellent; a systematic categorization of them would be very insightful.
3. Regarding the secondary evaluation with qwen3-max-preview, what was the approximate rejection rate? That is, what percentage of attacks that produced a different numerical answer (preliminary success) were later judged by the evaluator LLM to be based on a correct reasoning path for the perturbed question (and thus not a true attack success)?
4. Have you performed a qualitative analysis of these longer responses? Do they show the model expressing confusion, getting stuck in reasoning loops, exploring multiple contradictory paths, or simply being more verbose before failing? A few examples illustrating these behaviors would greatly enrich the paper.
5. The paper focuses on a single-word substitution attack. Did you explore other minimal perturbations, such as the insertion or deletion of a single, seemingly innocuous word (e.g., an article or adverb)? It would be interesting to know if the models are similarly sensitive to such structural changes.

---

> ### Author Response · Authors · 2025-11-20
>
> We sincerely thank you for taking the time to review our manuscript. Your valuable comments are critical for improving the quality of our paper. We greatly appreciate your insightful feedback. Below we provide detailed responses to the issues you raised, and we hope they address your concerns.
>
> **Question 1** : An inquiry about a more detailed breakdown of attack and the percentage of successful perturbations contributed by each of the three candidate sources.
>
> **Answer 1** : The following table provides a more detailed breakdown of the successful attacks, showing the percentage of successful perturbations originating from each of the three candidate sources (percentage, %).
>
> | Model   | GSM8K (Cosine) | GSM8K (WordNet) | GSM8K (MLM) | MATH500 (Cosine) | MATH500 (WordNet) | MATH500 (MLM) |
> | :----: | :-----: | :-----: | :-----: | :-----: | :-----: | :-----: |
> | DeepSeek-R1-Distill-Qwen-32B  | 60.35   | 15.37   | 24.28   | 57.48   | 12.18   | 30.34   |
> | DeepSeek-R1-Distill-Qwen-7B  | 60.90   | 16.69   | 22.41   | 46.68   | 16.06   | 37.26   |
> | DeepSeek-R1-Distill-Qwen-1.5B  | 70.69   | 9.87   | 19.44   | 60.25   | 8.70   | 31.05   |
> | Qwen2.5-Math-7B-Instruct  | 28.80   | 24.43   | 46.77   | 30.12   | 16.96   | 52.92   |
> | Qwen2.5-Math-1.5B-Instruct  | 49.61   | 18.75   | 31.64   | 50.50   | 15.64   | 33.83   |
> | Llama-3-70B-Instruct  | 33.61   | 13.30   | 53.09   | 59.08   | 3.34   | 37.58   |
> | Llama-3-8B-Instruct  | 61.30   | 16.02   | 22.68   | 53.32   | 2.01   | 44.67   |
> | gemma-3-27b-it  | 15.70   | 18.32   | 65.98   | 48.56   | 14.35   | 36.60   |
> | gemma-3-12b-it  | 18.36   | 17.89   | 63.75   | 46.13   | 13.77   | 40.10   |
> | gemma-3-4b-it  | 22.49   | 24.18   | 53.33   | 52.27   | 13.18   | 34.55   |
> | gemma-2-27b-it  | 17.36   | 19.32   | 63.32   | 43.68   | 9.42   | 46.90   |
> | gemma-2-9b-it  | 25.29   | 20.30   | 54.41   | 53.30   | 10.24   | 36.46   |
>
> Based on the results on GSM8K and MATH, we find that the MLM information source contributes the most to attack success, and in most models it clearly outperforms the other two candidate sources. At the same time, cosine similarity also performs strongly on some models, especially the DeepSeek series, which shows that semantically similar candidate words can also effectively induce incorrect reasoning. Overall, MLM candidates achieve higher success rates, suggesting that generative LLMs are more sensitive to candidate words predicted by language models. The different impacts of MLM, cosine similarity, and WordNet across models further indicate that different types of semantic similarity affect model robustness in different ways.
>
> **Question 2** : An inquiry about a systematic categorization of the types of perturbed words.
>
> **Answer 2** : That's a great question! Below is a table where we categorize the types of perturbed words in more detail, describing the proportion of each type, which will help us better understand the attack mechanisms (percentage, %).
>
> | Model     | PROPER_NOUN_ENTITY | COMMON_NOUN | VERB   | PREPOSITION | ARTICLE | ADVERB | ADJECTIVE | PRONOUN | OTHER  |
> |-----------|--------------------:|------------:|-------:|------------:|-------------------:|-------:|----------:|--------:|-------:|
> | DeepSeek-R1-Distill-Qwen-32B  | 18.7 | 23.5 | 20.9 | 10.0 | 7.6 | 4.6 | 5.6 | 4.3 | 4.8 |
> | DeepSeek-R1-Distill-Qwen-7B  | 24.3 | 19.9 | 22.8 | 8.3 | 9.9 | 1.5 | 5.3 | 3.9 | 4.1 |
> | DeepSeek-R1-Distill-Qwen-1.5B  | 42.0 | 12.7 | 19.0 | 9.7 | 9.3 | 1.6 | 1.6 | 3.3 | 0.7 |
> | Qwen2.5-Math-7B-Instruct  | 8.3 | 33.7 | 13.5 | 8.7 | 7.6 | 10.5 | 7.1 | 5.6 | 5.0 |
> | Qwen2.5-Math-1.5B-Instruct  | 18.4 | 26.7 | 17.4 | 9.1 | 5.0 | 8.3 | 5.7 | 6.9 | 2.4 |
> | Llama-3-70B-Instruct  | 12.7 | 28.7 | 13.5 | 7.1 | 6.5 | 12.6 | 7.3 | 6.3 | 5.4 |
> | Llama-3-8B-Instruct  | 30.4 | 18.5 | 18.7 | 10.5 | 8.8 | 4.2 | 2.6 | 4.8 | 1.5 |
> | gemma-3-27b-it  | 6.5 | 35.4 | 18.1 | 7.0 | 2.7 | 13.6 | 8.6 | 3.5 | 4.5 |
> | gemma-3-12b-it  | 8.9 | 33.1 | 16.8 | 5.9 | 4.3 | 10.5 | 10.8 | 4.9 | 4.7 |
> | gemma-3-4b-it  | 14.6 | 28.2 | 16.6 | 6.1 | 4.6 | 12.5 | 9.5 | 4.5 | 3.4 |
> | gemma-2-27b-it  | 14.1 | 29.2 | 14.4 | 7.1 | 5.3 | 9.1 | 14.1 | 3.5 | 3.1 |
> | gemma-2-9b-it  | 16.9 | 32.3 | 16.4 | 5.1 | 5.0 | 8.4 | 7.5 | 5.2 | 3.2 |
>
> As shown in the table, the perturbed words are often concentrated in three categories: proper nouns/named entities, common nouns, and verbs. These three categories account for more than 60% of the total, and in some models, the proportion even exceeds 70%. Additionally, we observe that for models within the same family, smaller models are more vulnerable to attacks involving proper nouns, which require external knowledge. However, as the model size increases, the perturbation of common nouns becomes more likely to successfully attack the model. This suggests that smaller models have more significant flaws in mathematical reasoning compared to larger models. While larger models show some improvement, they still have considerable issues.

---

> > ### Author Response · Authors · 2025-11-20
> >
> > **Question 3** : An inquiry about the rejection rate.
> >
> > **Answer 3** : We present the concept of the rejection rate you mentioned in the table below (percentage, %):
> >
> > | Model   | GSM8K | MATH500 |
> > | :----: | :-----: | :-----: |
> > | DeepSeek-R1-Distill-Qwen-32B  | 1.97   | 1.20   |
> > | DeepSeek-R1-Distill-Qwen-7B  | 1.06   | 0.40   |
> > | DeepSeek-R1-Distill-Qwen-1.5B  | 0.15   | 0.00   |
> > | Qwen2.5-Math-7B-Instruct  | 1.52   | 0.60   |
> > | Qwen2.5-Math-1.5B-Instruct  | 0.83   | 0.20   |
> > | Meta-Llama-3-70B-Instruct  | 1.74   | 0.60   |
> > | Meta-Llama-3-8B-Instruct  | 0.15   | 0.00   |
> > | gemma-3-27b-it  | 1.44   | 0.80   |
> > | gemma-3-12b-it  | 0.76   | 0.40   |
> > | gemma-3-4b-it  | 0.23   | 0.20   |
> > | gemma-2-27b-it  | 1.29   | 0.40   |
> > | gemma-2-9b-it  | 0.45   | 0.00   |
> >
> > As shown in the table, the approximate rejection rate for the secondary evaluation using qwen3-max-preview is very low. The purpose of using qwen3-max-preview for secondary evaluation is to make our evaluation mechanism more robust and to avoid minor accuracy discrepancies due to issues like unit conversion or cases where the model fails to put the answer in \boxed{}.

---

> > > ### Author Response · Authors · 2025-11-20
> > >
> > > **Question 4** : An inquiry about the qualitative analysis of overly long responses.
> > >
> > > **Answer 4** : We have performed a qualitative analysis of these longer responses, as suggested in your question, and have provided two additional examples for analysis. For non-thought models, they simply become more verbose before failing, as shown in the following examples:
> > >
> > > Original question:
> > >
> > > *Carla is downloading a 200 GB file. Normally she can download 2 GB/minute, but 40% of the way through the download, Windows forces a restart to install updates, which takes 20 minutes. Then Carla has to restart the download from the beginning. How load does it take to download the file?*
> > >
> > > Model's response:
> > >
> > > *Here's how to solve the problem step-by-step:*
> > >
> > > *Calculate the amount of data downloaded before the restart: 200 GB × 40% = 80 GB*
> > >
> > > *Calculate the time spent downloading before the restart: 80 GB / 2 GB/minute = 40 minutes*
> > >
> > > *Calculate the time lost due to the restart: 20 minutes*
> > >
> > > *Calculate the time for the final download from the beginning: 200 GB / 2 GB/minute = 100 minutes*
> > >
> > > *Calculate the total time: 40 minutes + 20 minutes + 100 minutes = 160 minutes*
> > >
> > > *\boxed{160} minutes*
> > >
> > >
> > > The model's response when "Carla" is replaced with "she"：
> > >
> > > *Here's how to solve the problem step-by-step:*
> > >
> > > *Step 1: Calculate the download time before the restart. The file size is 200 GB. The download speed is 2 GB/minute. Download time = File size / Download speed = 200 GB / 2 GB/minute = 100 minutes*
> > >
> > > *Step 2: Calculate the download progress at the time of the restart. The restart happens 40% of the way through the download. Download progress = 40% * 100 minutes = 40 minutes*
> > >
> > > *Step 3: Calculate the amount of data downloaded before the restart. Download speed = 2 GB/minute. Data downloaded = Download speed * Download time = 2 GB/minute * 40 minutes = 80 GB*
> > >
> > > *Step 4: Calculate the remaining download time. Remaining file size = 200 GB - 80 GB = 120 GB. Remaining download time = Remaining file size / Download speed = 120 GB / 2 GB/minute = 60 minutes*
> > >
> > > *Step 5: Calculate the total download time. Total download time = Download time before restart + Restart time + Remaining download time. Total download time = 100 minutes + 20 minutes + 60 minutes = 180 minutes*
> > >
> > > *Final answer: \boxed{180 minutes}*
> > >
> > > However, for models with explicit chain-of-thought capabilities, they often show clear signs of confusion or get stuck in reasoning loops. When the question is slightly perturbed, DeepSeek-R1-Distill-Qwen-1.5B may produce a very unintelligible reasoning process that keeps looping and ultimately fails to give a reasonable final answer, as shown in the following example. Due to the character limit, we only show part of the model’s response to the perturbed question here. In fact, this segment is essentially the model’s entire answer, because we observe that it keeps repeating this part without providing any meaningful reasoning or conclusion. Therefore, we only include this portion of the output.
> > >
> > > Original question:
> > >
> > > *Eliza's rate per hour for the first 40 hours she works each week is $10. She also receives an overtime pay of 1.2 times her regular hourly rate. If Eliza worked for 45 hours this week, how much are her earnings for this week?*
> > >
> > > The model's response when "the" is replaced with "THE":
> > >
> > > *To answer this question, Eliza needs to know:*
> > >
> > > *A) The regular hourly rate for THE first 40 hours.*
> > >
> > > *B) The overtime pay multiplier.*
> > >
> > > *C) The total number of hours she worked.*
> > >
> > > *D) The total number of hours she worked beyond the first 40 hours.*
> > >
> > > *Wait, this seems like a repeated question. Let me try to figure it out step by step.*
> > >
> > > *First, Eliza works 45 hours this week. The first 40 hours are at her regular rate of $10 per hour. The remaining 5 hours are considered overtime, and she gets paid 1.2 times her regular rate for those.*
> > >
> > > *So, to calculate her total earnings, she needs to know: Her regular hourly rate for the first 40 hours. Without this, she can't compute the base pay. The overtime pay multiplier, which is 1.2 times the regular rate. If she doesn't know this, she can't compute the overtime pay. The total number of hours she worked, which is 45. This is given, but maybe she can compute the overtime hours (45 − 40 = 5) without knowing the total hours.*
> > >
> > > *Wait, the question is asking what Eliza needs to know to calculate her earnings. So, if she knows her regular hourly rate, the overtime multiplier, and the total hours, she can compute her pay.*

---

> > > > ### Author Response · Authors · 2025-11-20
> > > >
> > > > **Question 5**: An inquiry about experiments involving the insertion of a single word.
> > > >
> > > > **Answer 5**: Your idea is very interesting! We are also curious about this question, so we run a preliminary experiment where we randomly insert a single, seemingly innocuous word into the original question (for example, an article, preposition, or adverb) that does not change the original meaning of the problem. We test this on two models over both the GSM8K and MATH500 datasets, and the results are shown in the table below (accuracy, %).
> > > >
> > > > | Model    | GSM8K (Original) | GSM8K(Attack) | MATH500 (Original) | MATH500 (Attack) |
> > > > |----------|----------:|----------:|---------:|---------:|
> > > > | Qwen2.5-Math-1.5B-Instruct | 85.90     | 83.93     | 76.20    | 74.40    |
> > > > | DeepSeek-R1-Distill-Qwen-1.5B | 83.09     | 80.14     | 81.20    | 80.20    |
> > > >
> > > > As the results show, when we only insert a single, seemingly innocuous word into the original question that does not change its intended meaning, the model’s performance slightly decreases, but the drop is not large. This suggests that LLMs are somewhat sensitive to this type of insertion-based minimal perturbation, but the sensitivity is not very strong and only leads to a mild degradation in accuracy. This, in turn, highlights the advantage of our single-word substitution attack, which is much more effective for probing the robustness of mathematical reasoning in LLMs.

---

> > > > > ### Author Response · Authors · 2025-11-27
> > > > >
> > > > > Dear Reviewer,
> > > > >
> > > > > We hope this message has not bothered you. With the discussion period ending soon, less than a week away, we would like to know if we have satisfactorily addressed the issues you raised. If you have any further questions, please let us know. Your feedback is invaluable to us, and we are eager to resolve your concerns to improve our work.
> > > > >
> > > > > Thank you again for taking the time to read our paper and response.

---

### Meta-Review · Area_Chair_5JXW · 2026-01-03

**Summary:**

This paper proposed an automated adversarial attack for robustness evaluation of LLM reasoning on mathematical tasks. There were several concerns raised on this paper. Beyond the minor concerns and clarification questions, the main concerns are:

- Multiple reviewers are concerned about the novelty, of both the conclusions that LLMs are sensitive to adversarial perturbations, and the attack framework.
- Multiple reviewers complained about the lack of detailed ablation studies, including studies of the contribution from each component of the framework, and multi-word attacks.

**Reviewer Concerns:**

The authors added ablation experiments for two-word attacks, and ablation studies showing the percentage of successful perturbations originating from each of the three candidate sources. The authors also provided detailed clarification to compare the relation to previous work.

**Reviewer Scores:**

I expect the reviewers originally voting for rejection to slightly increase their scores, and the reviewers originally voting for acceptance to keep their score. I think small technical questions are mostly clarified, and the main debate would lie on how much novelty or contributions each reviewer believe this paper brings to the field.

---

### Decision · Program_Chairs · 2026-01-26

Accept (Poster)